# NOISE TRANSFORMS FEED-FORWARD NETWORKS INTO SPARSE CODING NETWORKS

## ABSTRACT

A hallmark of biological neural networks, which distinguishes them from their artificial counterparts, is the high degree of sparsity in their activations. Here, we show that by simply injecting symmetric, random, noise during training in reconstruction or classification tasks, artificial neural networks with ReLU activation functions eliminate this difference; the neurons converge to a sparse coding solution where only a small fraction are active for any input. The resulting network learns receptive fields like those of primary visual cortex and remains sparse even when noise is removed in later stages of learning.

## 1  INTRODUCTION

The brain is highly sparse with an estimated 15% of neurons firing at any given time (Attwell & Laughlin, 2001). The most immediate answer for why is metabolic efficiency: action potentials consume ~20% of the brain's energy (Sterling & Laughlin, 2015; Attwell & Laughlin, 2001; Sengupta et al., 2010). However, there are further advantages to sparsity in the brain (Olshausen & Field, 2004). One significant advantage is improving the signal to noise ratio (SNR) of neural signals. Sparsity improves SNR by (i) turning off any weakly firing neurons activated by noise; (ii) increasing the separability of data points (Ahmad & Scheinkman, 2019; Xie et al., 2022).

Inhibitory interneurons that suppress all but the most active neurons from firing are an important mechanism for enforcing this sparsity (Haider et al., 2010). Theoretical and empirical results support their involvement in both silencing noise and separating neural representations. Examples include horizontal interneurons in the retina (Sterling & Laughlin, 2015) and Golgi interneurons in cerebellar-like structures (Fleming et al., 2022; Lin et al., 2014; Xie et al., 2022).

Biologically, as depicted in Fig. 1, these inhibitory interneurons implement a negative feedback loop whereby the more active the excitatory neurons are, the more active the interneuron becomes, and hence the more it inhibits these excitatory neurons. Simplified models of the circuit written as ordinary differential equations (ODEs) show convergence to an approximate $k$ of the most active neurons remaining on (Gozel & Gerstner, 2021). This will be referred to as a Top-K activation function (also known as k Winners Take All). There is empirical support for a number of interneuron circuits approximating the Top-K operation (Sterling & Laughlin, 2015; Fleming et al., 2022; Lin et al., 2014).

By contrast, in the field of deep learning, while inhibition is possible, analogous interneuron circuits that enforce sparsity across a layer have not been widely adopted. The only truly sparse activation function is the ReLU (Glorot et al., 2011). Moreover, mechanisms that enforce sparse neuronal activity are also rarely used and when given the choice, networks will prefer to be dense. This is because sparsity can limit model capacity, resulting in information bottlenecks that harm performance (Goodfellow et al., 2015).

Here, we find that by simply introducing isotropic, symmetric noise centered about zero during training, a layer of artificial neurons will converge to a sparse coding solution. This solution mimics a simplified version of the biological inhibitory interneuron circuit. The network gradually implementing this inhibitory interneuron also results in better performance than explicitly enforcing any sort of inhibition at the start of training.

Concretely, the network synchronizes every neuron's bias term, setting them to be approximately the same negative value, and also every neuron's weight vector, setting them to have the same $L_2$ norm.

This results in the activity of every neuron existing within a particular range that, when combined with the negative bias term they all agree upon, results in a sparse $k$ neurons remaining active.

Because we assume that $k$ is small, Top-K networks are a special class of sparse coding networks that always have an approximate $k$ neurons on for any input. They also implement this sparsity by approximating the functionality of an inhibitory interneuron. The sparsity that comes from inhibition – allowing a small subset of neurons to fire from the total number that otherwise would – is distinct from sparsity where many of the neurons in the network are *dead* and never fire for any input. This latter form of sparsity is misleading and equivalent to having a pruned, smaller network that is densely firing; we indicate when this alternative form of sparsity occurs.

The degree to which our network becomes a sparse coding, Top-K network is proportional to the amount of noise applied, up to a noise limit. To validate that the network approximates an inhibitory interneuron, we replace each neuron's bias term with a single, shared bias term. This results in identical model performance and very similar levels of sparsity. Further investigating the degree to which this network uses sparse coding, we find that it learns receptive fields similar to those of mammalian V1 with Gabor filters and retinal ganglion cells with on/off center surround (Sterling & Laughlin, 2015; Olshausen & Field, 1997).

We find this Top-K network formation is particularly evident for reconstruction tasks but also exists for classification tasks and the intermediate MLP layers of a Transformer architecture. Our results hold across a variety of datasets, noise distributions, and numbers of neurons. We also observe that the network retains its Top-K approximation even after the noise is removed, making it an effective pre-training task for sparsifying activations. This increase in sparsity could reduce FLOPs when run on hardware that can take advantage of it (Wang, 2020; Gale et al., 2020; Davies et al., 2018).

We first review related work (Section 2) before outlining our experimental setup and presenting empirical observations (Section 3). Finally, we analyze the network's learning dynamics, providing intuition for our results and highlighting avenues for future work (Section 4).

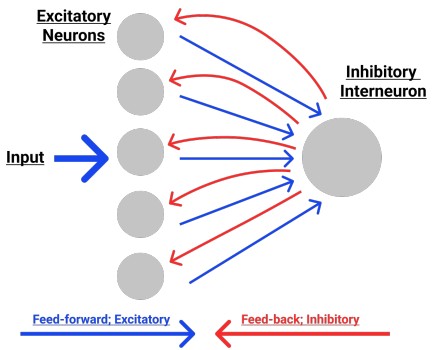

Figure 1: An inhibitory interneuron circuit implementing a negative feedback loop to silence all but the $k$ most active neurons.

## 2 RELATED WORK

Injecting noise into training data for statistical models was previously proposed by (Sietsma & Dow, 1991). Gaussian noise injection has been interpreted as a form of model regularization, to help avoid overfitting (Zur et al., 2009) and improve generalization (Sietsma & Dow, 1991; Matsuoka, 1992). It was later shown that noise injection is in fact equivalent to having a regularization term that seeks to minimize the $L_2$ norm of the network's Jacobian (Bishop, 1995; Rifai et al., 2011; Alain & Bengio, 2014). From this perspective, it makes sense that sparsity can be used to turn off neurons and minimize this Jacobian, providing a potential explanation for our results.

Training with noise became particularly prominent in the form of de-noising autoencoders (Goodfellow et al., 2015). Before deep neural networks with many layers could be trained end-to-end, it was popular to train each layer of the network one at a time with an unsupervised, de-noising reconstruction loss (Goodfellow et al., 2015). With small amounts of noise, these de-noising autoencoders were shown to be equivalent to contractive auto-encoders that were regularized to be robust to local perturbations of the training data (Alain & Bengio, 2014).

Approaches to introduce activation sparsity have included explicitly using Top-K activation functions (Ranzato et al., 2007; Makhzani & Frey, 2014; Ahmad & Scheinkman, 2019), novel regularization terms (Kurtz et al., 2020; Yang et al., 2020) and other approaches (Schwarz et al., 2021; Molchanov et al., 2017). However, we are unaware of any existing work which finds that noisy

training with a ReLU activation function not only results in sparsity but also converges to a sparse coding, Top-K network.

Learning reconstructions of training data with sparse activations is the mainstay of sparse coding (Olshausen & Field, 1997). Sparse coding uses an overcomplete basis to learn a reconstruction of data with an $L_1$ regularization penalty on neuron activity. The image reconstruction can be written as $\hat{\mathbf{x}} = \sum_i^N a_i \phi_i$ where $\phi_i$ is a basis vector from the matrix or "dictionary" $\Phi$; $a_i$ is the activity of this basis vector, and the activation weighted sum of basis vectors produces the output $\hat{\mathbf{x}}$. The objective function aims to minimize the mean squared-error between the reconstructed image and the original image while having sparse neuron activity: $\arg\min_{\Phi, \mathbf{a}} (\mathbf{x} - \hat{\mathbf{x}})^2 + \lambda \sum_i^N |a_i|$. This sparse coding task is an example of a statistical model converging with neuroscience because, in addition to producing sparse representations, the basis functions learn to become Gabor filters like those found in area V1 of the visual system (Olshausen & Field, 1997).

Interestingly, the ReLU activation function that is crucial for the results of this paper can be derived as the optimal solution to a simplified sparse coding problem when we assume that the basis vectors are orthogonal (Ba, 2020; Gregor & LeCun, 2010). This results in the optimal solution for activity $a_i$ to be given by $\text{ReLU}(u_i - \lambda) - \text{ReLU}(-u_i - \lambda)$ or, if we constrain $a_i$ to be non-negative, $\text{ReLU}(u_i - \lambda)$, where $u_i$ is the value obtained by projecting the data onto basis $\phi_i$. Intuitively, if the values of our target image $\mathbf{x}$ reconstructed by $a_i \phi_i$ are too small, it is better for $a_i = 0$ than to predict $\mathbf{x}$, where the threshold for this tradeoff is given by $\lambda$, which behaves as an activation threshold.

Top-K activation functions inspired by inhibitory interneurons have been implemented in a number of deep learning models for greater computational efficiency (Ahmad & Scheinkman, 2019), adversarial robustness (Xiao et al., 2020; Tramèr et al., 2020), interpretability (Nelson et al., 2022), and continual learning (Shen et al., 2021; Iyer et al., 2022). However, in all of these instances there are explicit mechanisms built into the network to enforce Top-K and handle repercussions such as creating dead neurons.

These Top-K networks, when trained with noise, are also closely related to associative memory models such as Sparse Distributed Memory (SDM) and Modern Hopfield Networks that activate a sparse subset of neurons and were developed to handle noisy queries (Kanerva, 1988; Krotov & Hopfield, 2016). SDM in particular can be viewed as a de-noising autoencoder and has been developed into an MLP that uses a Top-K activation function (Keeler, 1988). Finally, de-noising autoencoders are single step versions of diffusion models (Song & Ermon, 2019).

## 3 RESULTS

**Experimental Setup -** By default, we train a single hidden layer ReLU network with different amounts of random Gaussian noise applied to the input. The objective is to remove the noise and reconstruct the original input. Formally, noise is applied from an isotropic Gaussian with variance $\sigma$ to the input $\mathbf{x}$. The corrupted image is then put through our encoder and decoder. For the single hidden layer case of Eq. 1 the encoder and decoder weight matrices and bias terms are: $W_e \in \mathbb{R}^{m \times n}, \mathbf{b}_e \in \mathbb{R}^m, W_d \in \mathbb{R}^{o \times m}, \mathbf{b}_d \in \mathbb{R}^o$ where $n$ is the input dimension, $o$ is output dimension, and $m$ is the number of neurons in the hidden layer. We will use the term "key vector" to refer to rows of $W_e$ and "value vector" to refer to the rows of $W_d$ that correspond to each neuron. Our loss function uses the mean squared error between the original image and reconstruction across our full dataset, $X$.

$$\tilde{\mathbf{x}} = \mathbf{x} + \epsilon, \ \ \epsilon \sim N(0, \sigma I)$$
$$\hat{\mathbf{x}} = W_d \, \text{ReLU}(W_e \tilde{\mathbf{x}} + \mathbf{b}_e) + \mathbf{b}_d$$
$$\text{loss} = \sum_{\mathbf{x} \in X} (\mathbf{x} - \hat{\mathbf{x}})^2. \tag{1}$$

We use Kaiming randomly initialized weights (He et al., 2015) and train until the fraction of active neurons converges. We primarily use the CIFAR10 dataset of 50,000 images of 32x32x3 dimensions, training either on the raw pixels (flattening them into a 3072 dimensional vector) or latent embeddings of 256 dimensions, produced by a ConvMixer pretrained on ImageNet (Trockman & Kolter, 2022; Russakovsky et al., 2015).

To ensure our results are robust, we vary a number of settings, finding that in every case the relationship between noise and network sparsity holds. All code and training parameters can be found at: `https://github.com/anon8371/AnonPaper2`.

**Symmetric Mean Zero Noise Induces Top-K Networks -** Figure 2 shows that a single hidden layer ReLU network with 10,000 neurons learns to become a Top-K network with smaller $k$ values as the amount of noise increases. This network was trained on the CIFAR10 embeddings to mimick training within a deeper network layer and where the manifold is smoother than when training on raw image pixels.

While Fig. 2 only shows the mean number of neurons that are active across all 50,000 CIFAR10 inputs per epoch, Figure 3 shows the fraction of active neurons for every input. This reveals that as noise increases, the variance for how many neurons are active shrinks, so that the mean is an accurate summary statistic. Note that there are no dead neurons. Fig. 3 also shows the bias terms for each neuron and the $L_2$ norm of their key vectors. It is clear that these values all synchronize to a value decided on by the network and are responsible for the sparse $k$ neurons remaining active after the ReLU nonlinearity is applied.

As noise increases, the bias terms become more negative to raise the activation threshold and increase sparsity. The $L_2$ norms of the key vectors also become smaller. We hypothesize this is a counter-response to the $L_2$ norms of the data increasing with noise and explains why in high noise settings ($\sigma \geq 0.8$) neuron activations linger around $\sim 50\%$ sparsity before rapidly becoming very sparse (see Fig. 2). Before the key vector $L_2$ norms are sufficiently small, noisy data with large $L_2$ norms produce very large neuron activations (both positive and negative) and the bias terms initialized close to 0 have no real effect on which neurons are active. Because the key vectors are randomly initialized from a symmetric distribution around 0, in expectation half of the neurons are active after the ReLU nonlinearity is applied. This means the network must wait to first shrink its key vector $L_2$ norms before the bias terms can affect activation sparsity.

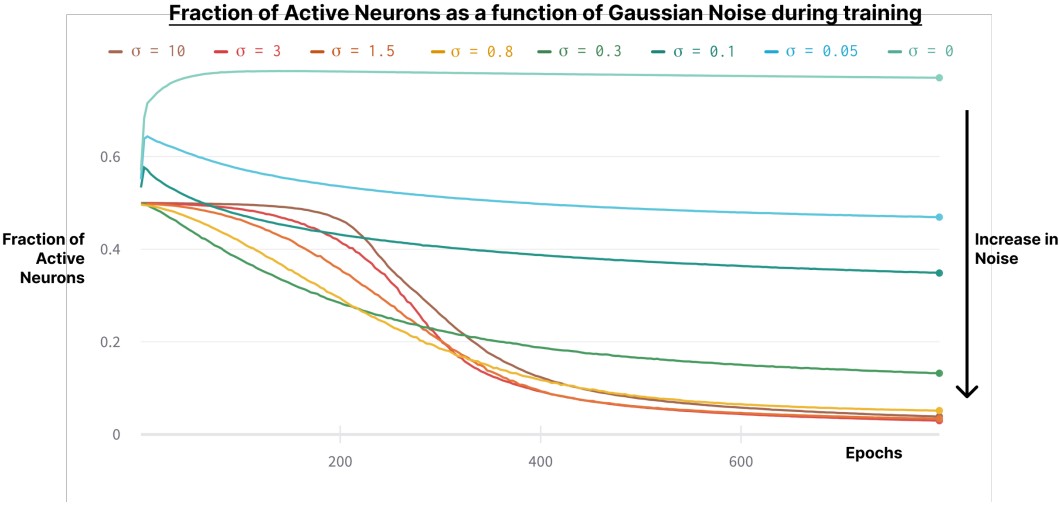

Figure 2: **There is a positive relationship between activation sparsity and noise.** The average fraction of neurons active during each CIFAR10 latent input over the course of training for 800 epochs. Each line corresponds to training randomly initialized networks with different noise levels $\sigma$ denoted by different colors. We show the average of three different random seeds and their standard error of the mean (not visible as the variance is so low). Interestingly, the higher noise levels take longer to sparsify but upon convergence become the sparsest.

To gain more intuition for the amounts of noise applied and reconstruction quality, we train networks directly on CIFAR10 pixels and demonstrate their results in Fig. 4. Unsurprisingly, the more noise that is applied, the worse reconstruction performance is. This is especially for noise levels large enough to make a competing image closer to the noisy input than the noise-free target image.

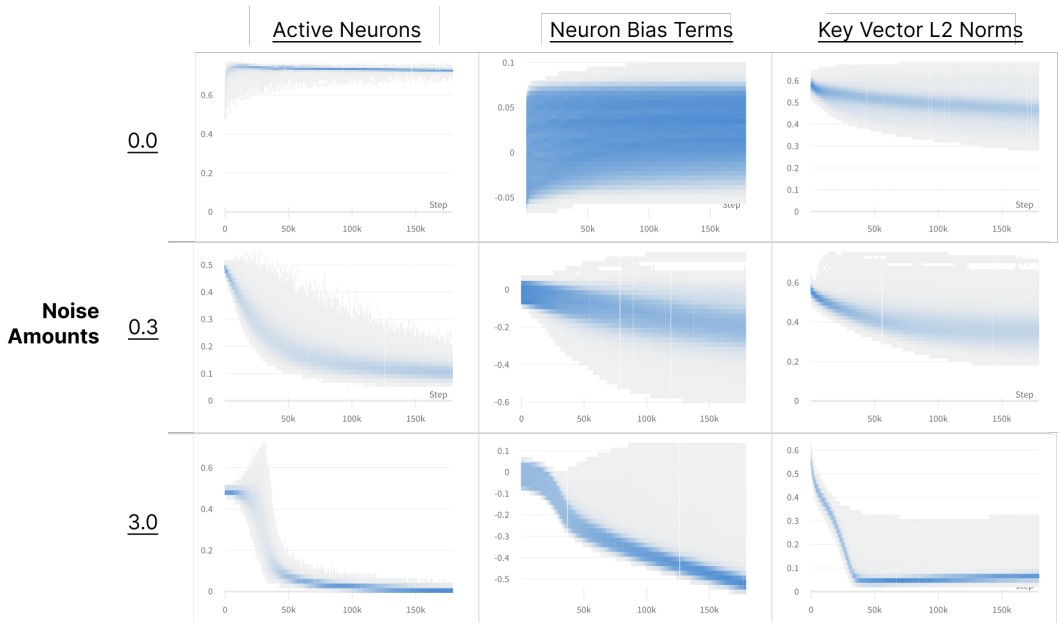

Figure 3: **Noise induces the formation of Top-K networks.** Shown column-wise are the fraction of neurons that are active for each training input (Left), the values of each of the 10,000 neurons bias terms (Middle) and the $L_2$ norms of their key vectors (Right), as a function of number of training epochs. Each is shown for three different representative noise levels (rows). These plots show each input to the network and use the density of blue to represent how many times a particular value occurs.

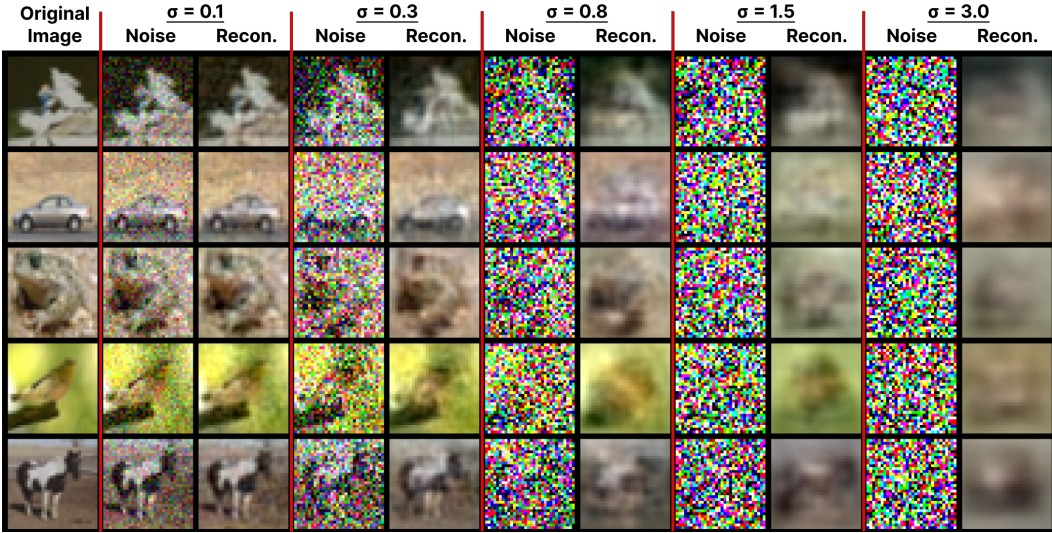

Figure 4: **Example CIFAR10 reconstructions.** Shown are example reconstructions obtained at different noise levels for five randomly selected images from the test data. The network at $\sigma \geq 0.8$ qualitatively transitions from fuzzy reconstructions to more general image details.

**Shared Bias Terms and Inhibitory Interneurons -** To test the observed transition into a Top-K network, we re-train our networks, replacing each neuron's bias term with a global shared bias term. This results in identical model performance and similar $k$ values. Using this shared bias term is interesting because, in conjunction with the synchronization of the $L_2$ weight norms, it is analogous to dynamically learning a $k$ value for the network.

We compared this network with explicitly creating sparse Top-K networks. These Top-K networks sort the neuron activations and keep only the $k$ most active. They were either given fixed $k$ values or could learn the $k$ values using either a continuous relaxation that is differentiable or the REIN-FORCE algorithm (see Appendix A for details) (Williams, 2004). These approaches all performed worse than the shared bias (and default ReLU) networks, with the best networks getting similar validation accuracies as the much smaller 100 neuron network and only when they were not sparse, using $k = 3,000$ neurons. Even the approaches learning $k$ were insufficiently dynamic and depended too much upon the $k$ initialization value.

**Biological Receptive Fields -** Looking more closely at the sparse solutions found by our ReLU networks, we discovered that for $\sigma \geq 0.3$, neurons begin to form receptive fields reminiscent of V1 Gabor filters and on/off center surround retinal ganglion cells (Sterling & Laughlin, 2015; Olshausen & Field, 1997). Figure 5 shows the receptive fields of the 425 neurons most activated (sorted from left-to-right and top-to-bottom) by a noisy image of a car when $\sigma = 0.8$. This is a similar result to that of sparse coding (Olshausen & Field, 1997), however, here we use noise with the ReLU activation function instead of a sparsity penalty and train on full images instead of 8x8 patches. See Appendix B for more receptive fields from models trained at every noise level.

The longer training progressed, the more similar the receptive fields appeared to those found in biology. These receptive fields are also unique to the noisy, reconstruction task. When we tried training the network without any noise but instead with an $L_1$ activation penalty motivated by the original sparse coding setup (Olshausen & Field, 1997), no biological receptive fields were observed. In addition, this approach resulted in unstable training, more dead neurons, less sparsity, and no synchronization of key vector norms or bias terms.[1]

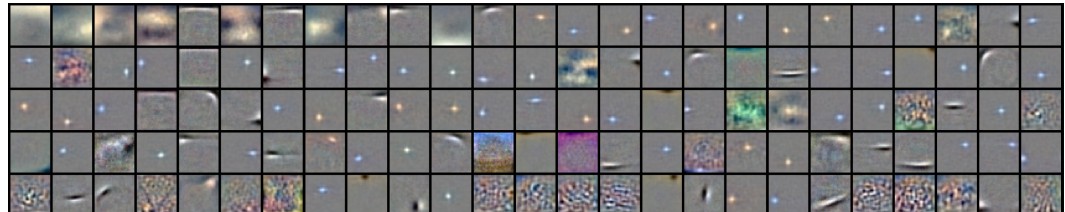

Figure 5: **Biological receptive fields form with noise.** Networks trained on CIFAR10 pixels with $\sigma = 0.8$ Gaussian noise. The most active 125 neuron receptive fields are reshaped into 32x32x3 and re-scaled so their values span the range of pixel values $[0, 255]$. Neurons are sorted by their activity levels (top left is the most active, going across rows then down columns) for a random noisy input of a car. Starting at $\sigma \geq 0.1$ and particularly for $\sigma = 0.8$ (shown here) we observe the formation of many Gabor filters and on/off center surround receptive fields that are unique to noisy training.

**Explaining Away and Facilitation -** Motivated by the similarity of our results with the sparse coding solution, we adopt more ideas from sparse coding, specifically the ability to "explain away" and "facilitate" (Olshausen & Field, 1997). Explaining away reduces the activity of neurons that are redundant with others that are already active, while facilitation encourages co-activation of neurons encoding a common structure such as a contour. We present our implementation and evaluation of these models in the Appendix C.

With these models we find that our networks become even more sparse and do so more quickly, as we would expect due to the explaining away of redundant neurons (see Appendix C). The network also converges faster and overfits to the training data. Using optimal stopping, we get the same performance with explaining away as with the best ReLU model. This suggests that the default ReLU network is already surprisingly effective at de-noising.

**Noisy Pre-Training Retains Model Sparsity without Performance Costs -** Investigating our noise trained sparse networks, we find that even after the noise is removed, they remain sparse. We train our models in the same way as in Fig. 2 but at epoch 800 linearly anneal each noise level down to 0.0 over the next 800 epochs. Fig. 6 shows that the sparsity levels remain largely unperturbed.

---

[1]We did not implement the full sparse coding framework, only the $L_1$ penalty, however, this shows that it is insufficient to implement sparse coding while noisy training is.

For classification, it is clear that while the correlation between noise and sparsity holds, it is not quite as strong. We believe this is due to the fact that classification only needs to cluster the data into the ten CIFAR10 labels, creating less interference between neural representations.

There is also no reduction in reconstruction or classification performance. In fact, for classification $\sigma \in \{0.1, 0.3\}$ both get 94.3% validation accuracy instead of the 93.4% that the baseline gets, showing slightly better generalization while also being more sparse (66% for baseline versus 55% for $\sigma = 0.1$ and 45% for $\sigma = 0.3$). Note that for the networks with $\sigma \geq 0.8$ for reconstruction and $\sigma \geq 3.0$ for classification, annealing the noise results in dead neurons, presumably because the data manifold no longer spans as large a space. However, these dead neurons do not harm task performance or entirely explain the sparsity levels. See Appendix D for data on the fractions of dead neurons.

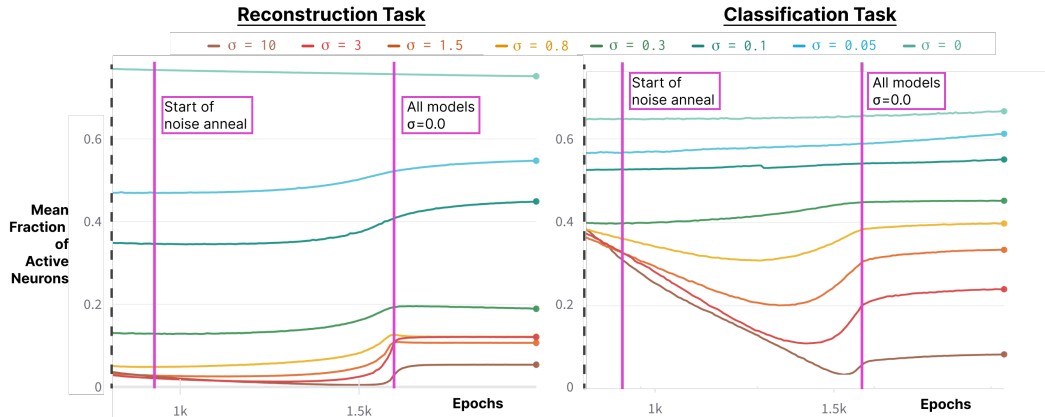

Figure 6: **Noisy pretraining retains network sparsity.** The vertical purple lines denote the start of noise annealing where each noise level is linearly decayed down to $\sigma = 0$. For both reconstruction (left) and classification (right) tasks the networks remain highly sparse even after all noise is removed. Moreover, there is no noticeable performance difference between any of the networks while some of those trained with moderate noise perform slightly better than the noise free baseline. We truncate the x-axis to start at epoch 600 for the sake of clarity. The classification networks continue to sparsify even when noise annealing has started because unlike in the reconstruction task, the sparsity level did not converge within the first 800 epochs.

**Model Ablations -** Figure 7 shows the mean neuron sparsity at network convergence across a number of different experimental conditions: (i) three datasets: CIFAR10 pixels, CIFAR10 latent embeddings, MNIST; (ii) two noise distributions: Gaussian and Laplace; (iii) four numbers of neurons: 100, 1,000, 10,000 and 100,000; (iv) three training tasks: reconstruction, classification, and Transformer next token prediction.

We ran hyperparameter sweeps of different learning rates for Adam, RMSProp, SGD, and SGD with Momentum (SGDM). We found that every optimizer with a high enough learning rate converged to a sparse Top-K solution without creating dead neurons.[2] We also tested the sigmoid and GELU activation functions. While neither of these functions is truly sparse in setting activations to 0, we counted neurons with absolute activation values $< 0.0001$ as "off". We found for the latent CIFAR10 dataset, both sigmoid and GELU (Hendrycks & Gimpel, 2016) had no sparsity but while sigmoid performs worse, GELU does just as well as ReLU. If we raised our arbitrary activation threshold to $< 0.01$ we found that GELU does become sparse with respect to noise but still not as sparse as ReLU while sigmoid remains fully dense (see Appendix E). Interestingly, for the raw CIFAR10 dataset, both sigmoid and GELU using the lower $< 0.0001$ threshold did become sparse and perform as well as ReLU. While still neither activation function becomes as sparse as ReLU, the GELU networks implement the same Top-K convergence as ReLU shown in Fig. 3 and form biologically plausible receptive fields like those of Fig. 5.

---

[2]Interestingly for SGD, with a low learning rate and enough epochs the model could learn non-sparse solutions with ~50% neuron activity and no updating of bias terms that performed almost as well as their sparse counterparts.

As part of ablating tasks, we test Transformer next token prediction. We train a small GPT2 models (Radford et al., 2019) with three blocks of interleaved Attention and Feedforward MLP layers. In front of each MLPs of each network, we inject one of the following noise levels $\sigma \in \{[0.0, 0.05, 0.1, 0.8, 1.5, 3.0, 8.0\}$ and use the WikiText-103 dataset. At global training step 200k, we remove all noise to also investigate the longer term pretraining effects.

The correlation between noise and activation sparsity remains and is again retained even after noise is removed. Furthermore, aside from the highest noise level of 8.0, pre-training has no effect on model performance. Figure 7(d) shows the final sparsity amounts of the first model layer with each of the other layers giving similar results with $\sim$3x to $\sim$9x more sparsity when $\sigma = 3.0$, compared to $\sigma = 0.0$.

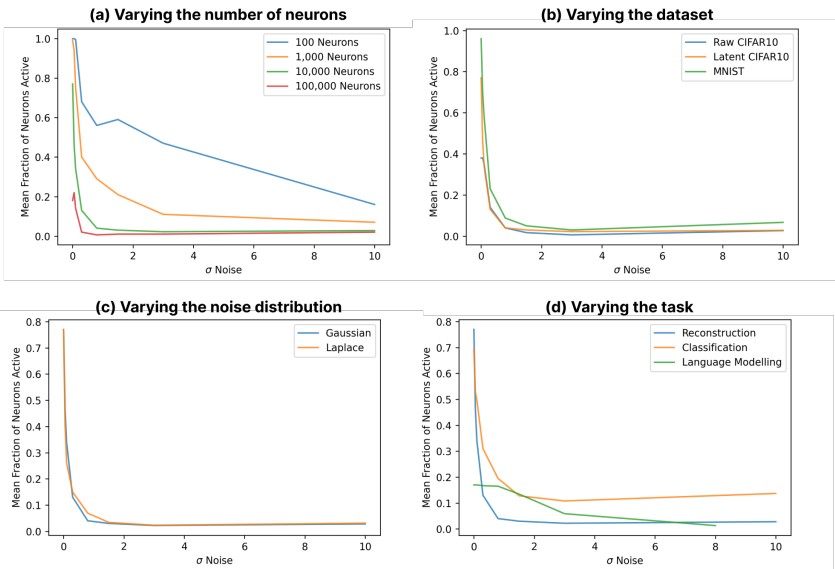

Figure 7: **Training ablations.** The positive correlation between noise and sparsity is robust to **(a)** neuron counts, **(b)** datasets, **(c)** noise distributions, and **(d)** training tasks.

## 4 DISCUSSION

Why does noisy training induce our ReLU network to become a biologically plausible Top-K network? Fundamentally, the network should want to use as many neurons as it can to give the best reconstruction of the input data as possible. However, it must trade-off the increase in accuracy gained by pooling more neurons with increased interference. This interference is caused by both noise and non-orthogonality of the learned weight vectors (Elhage et al., 2022).

An increase in noise will create more erroneous low-level activations in neurons that must be ignored to maximize signal to noise ratio. Therefore, during noisy training, the model must learn weight vectors to encode information about the data distribution that can be separated from the noise using only ReLU functions. In natural images, for example, certain projections (e.g., with Gabor filters) have high kurtosis which allows them to be easily distinguished from noise via thresholding or "coring" (Simoncelli & Adelson, 1996). Similarly, we hypothesize that the encoding layer in our network is learning such projections, so that thresholding performs noise rejection while also producing sparse activations. In theory, incorporating the "explaining away" of sparse coding networks should additionally remove interference due to the non-orthogonality of weight vectors. Our initial attempt at doing so did not produce performance gains, though it did increase sparsity. However further investigation is warranted to understand its potential benefit.

The positive correlation between noise and sparsity is opposite to Transformer Attention and associative memory models such as SDM and Hopfield Networks (Bricken & Pehlevan, 2021). For Transformer Attention, the difference is that Attention has direct access to every input within its

receptive field but no further. This provides for high accuracy when dealing with these inputs (keys) but an inability to store long term memories. As a result, in the low noise regime, it is optimal for Attention to implement a form of nearest neighbour lookup, activating few keys. When there is more noise, especially noise that moves the query so far from its target key that it is no longer the nearest, it is optimal to activate more keys and average over them, losing accuracy but giving a solution in the correct neighbourhood.

By contrast, SDM can store long term memories. However, it has the same negative correlation between noise and sparsity as Attention because it attempts to store full patterns with maximum fidelity within each neuron instead of distributing the features of each pattern across neurons (Kanerva, 1988). This results in SDM attempting to also perform a nearest neighbour lookup, activating only as many neurons around the noisy query as is necessary to activate those storing the target pattern.

We have shown the extent to which a feedforward ReLU MLP is capable of converging with a biological inhibitory interneuron circuit. Using deep learning to give insights into neuroscience, this result suggests that in the presence of noise, the biological sparse coding solution is optimal. Moreover, the fact that noise alone results in sparse coding rather than having a sparsity penalty in the loss function suggests that the primary driver behind sparse coding may be the handling of noise with metabolic efficiency as an added benefit, rather than the other way around. In addition, the fact that the reconstruction loss converges to this solution hints that the brain may be trying to maximize mutual information of its sensory stimuli (Sterling & Laughlin, 2015).

**Limitations -** Our work is largely empirical. An important goal of future work will be to develop a rigorous mathematical theory for modelling how the amount of noise affects sparsity for a given dataset, rather than having to test a number of noise levels to find one that works (analogous to tuning hyper-parameters).

In addition, while we show that our de-noising autoencoder converges with the biological sparse coding network, conventional hardware accelerators are unable to benefit from the increased levels of unstructured sparsity. Meanwhile, the advantages of noisy training for generalization, continual learning, and adversarial robustness have already been documented. However, sparse networks have recently been shown to aid mechanistic interpretability by removing polysemantism and encouraging specialization (Nelson et al., 2022) and additional future applications may arrive.

**Conclusion -** We have shown that the realistic assumption of noisy training data is sufficient for feed-forward layers to organically become highly sparse and retain their sparsity, even after noise is removed. This result is particularly striking because of the way the networks converge with sparse coding, producing biological receptive fields and approximating an inhibitory interneuron that implements a Top-K operation. Combining noisy training with other approaches to induce sparsity, including taking additional ideas from sparse coding, may result in even higher degrees of sparsity. More broadly, this work builds a new bridge between artificial and biological neural networks by showing how they can be more similar than they may otherwise appear.

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

# Appendix

## Table of Contents

## A  LEARNING TOP-K VALUES

There are two different approaches taken to learn the $k$ values in the explicit Top-K networks. The first is to use a continuous relaxation of the $k$ value that is otherwise a non-differentiable integer. We create a window of size $w$ and apply a sigmoid weighting to all values between $k \pm w$. We can then let $k$ be a continuous value that is responsible for shifting the center point of this sigmoidal activation window.

Another solution is to create a probability distribution over the possible $k$ values, sample discrete $k$ integers from it, and use the REINFORCE algorithm (Williams, 2004) to update the sufficient statistics of our distribution. Intuitively, we can think of this approach as trying a bunch of different $k$ values and updating to see more of those that lead to lower loss.

Both of these implementations can be found in our github repo at the following URL: `https://github.com/anon8371/AnonPaper2`.

## B  RECEPTIVE FIELDS

Here we show additional images of the receptive fields for networks trained with different amounts of noise (Fig. 8), the queries used to activate these neurons (Fig. 9) and how the receptive fields look more biological as the network is trained for longer (Fig. 10).

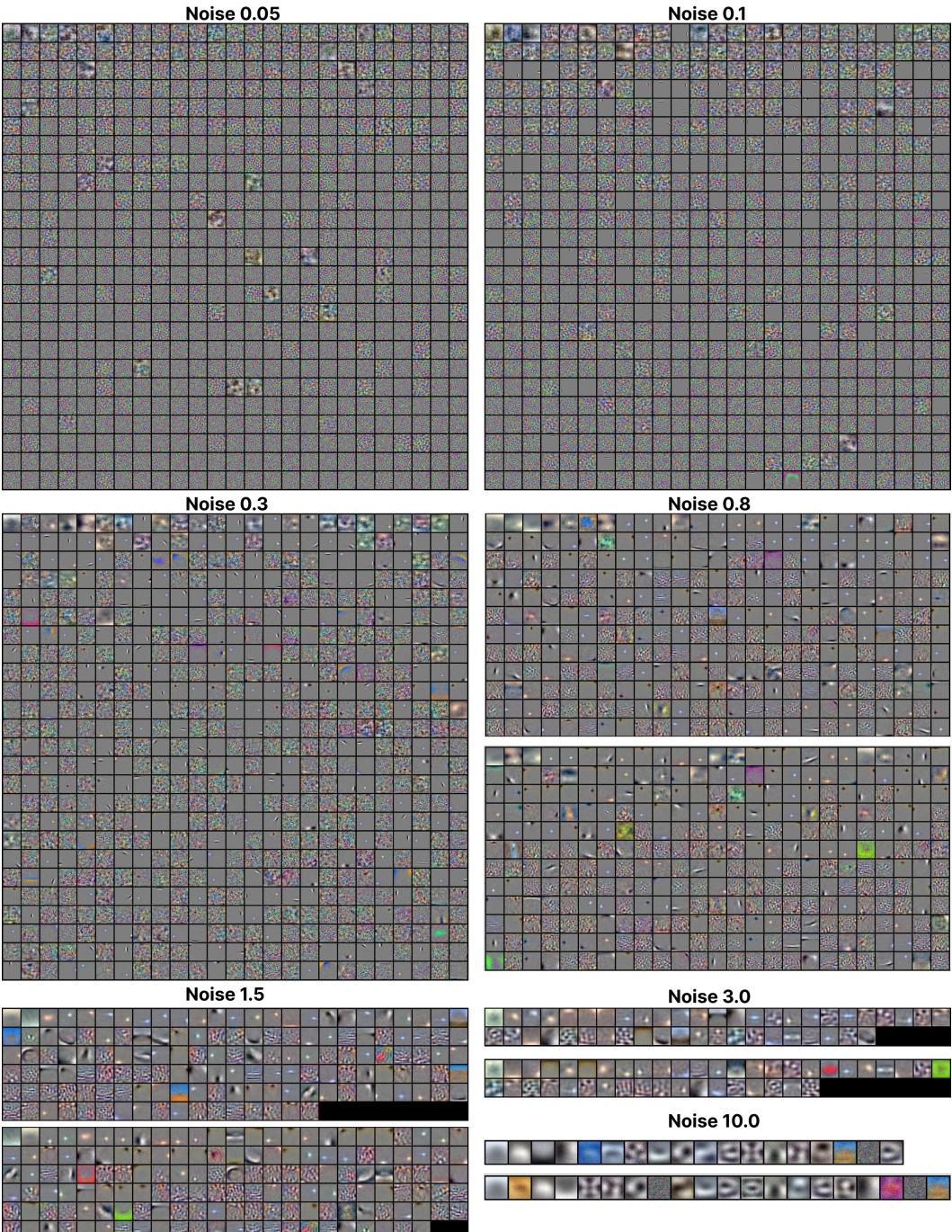

Figure 8: **Receptive fields across noise levels.** We show for all noise levels the 625 most active neurons for the truck query (middle of Fig. 9). For $0.8 \geq \sigma$ which activate fewer neurons, we also show the cat query below it. Biological like receptive fields start to appear for $\sigma = 0.1$ and can be seen most clearly for $\sigma \in \{0.3, 0.8, 1.5\}$.

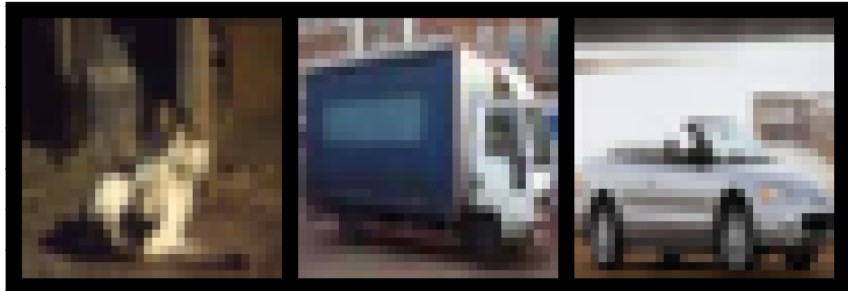

Figure 9: **Queries used to activate the receptive fields.** These queries were randomly selected. We use the cat (left) and truck images (middle) with the relevant noise levels to activate the neurons shown in Fig. 8. We use the car image (right) for the main text Fig. 5 and receptive field changes across epochs in Fig. 10.

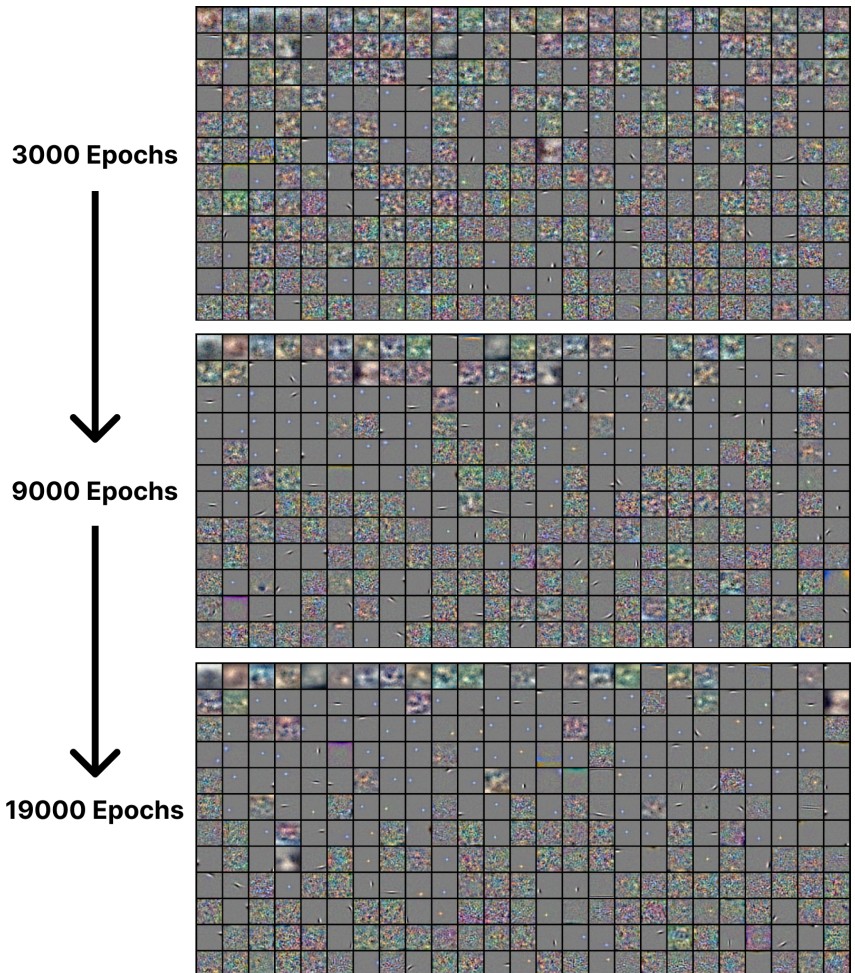

Figure 10: **Receptive fields become more biological across epochs.** We show how the most active receptive fields for the $\sigma = 0.3$ network evolve over the course of training to become more biologically similar. The specific number of epochs used (3,000, 9,000 and 19,000) were chosen simply because of when the models were checkpointed and re-loaded for additional training. We use the car image as our noisy query here (rightmost in Fig. 9).

## C    EXPLAIN AWAY

We can model explaining away and facilitation with an additional weight matrix $G \in \mathbb{R}^{m \times m}$ (recall $m$ is the number of neurons) that we run in a recurrent inner loop. Our one hidden layer network becomes:

$$
\begin{aligned}
\mu_0 &= W_e \tilde{\mathbf{x}} + \mathbf{b}_e \\
\mathbf{z}_t &= \text{ReLU}(\mu_t) \\
\Delta\mu &= \left(\mu_0 - G\mathbf{z}_t\right) - \mu_t \\
\mu_{t+1} &= \mu_t + \gamma\Delta\mu \\
\hat{\mathbf{x}} &= W_d \, \text{ReLU}(\mu_{t^*}) + \mathbf{b}_d,
\end{aligned}
\tag{2}
$$

where $t$ increments from 0 to $t^*$ as the number of iterations we want the inner loop to optimize neural activity and $\gamma$ is the size of the update step. In the simplest case where $t^* = 1$ and $\gamma = 1$, we do a single forward pass through this circuit making it analogous to a residual connection that comes before the activation function. More generally, the value of $t^*$ can be thought of as the number of recurrent connections that all originate from $\mu_0$ and end after every update to $\mathbf{z}$. The original Eq. 1 is a special case where $t^* = 0$. This use of $G$ learnt with backpropagation was first explored in Gregor & LeCun (2010).

Training the same networks as before but using Eq. 2 with $t^* = 1$ and $\gamma = 1$ we present our results in Table 1. We show the sparsity amounts of the networks in Fig. 11.

We can also replace $G\mathbf{z}_t$ with $G_d G_e \mathbf{z}_t$ where $G_e \in \mathbb{R}^{1 \times m}, G_d \in \mathbb{R}^{m \times 1}$ implement a single inhibitory interneuron that uses $2m$ rather than $m^2$ parameters and must learn each of its afferent and efferent connections. When the neurons still have their bias terms this solution has no effect. Removing the bias terms and relying just on the inhibitory interneuron, we found that it performs well (Table 1) and becomes sparse as a function of noise but results in a number of dead neurons and is not as sparse as the baseline ReLU or Explaining Away models. However, this shows that the full inhibitory interneuron solution is at least plausible.

| Method — Val. Acc. | $\sigma = 0.05$ | $\sigma = 0.1$ | $\sigma = 0.3$ | $\sigma = 0.8$ |
|---|---|---|---|---|
| ReLU $m =$100 | 0.0047 | 0.0067 | 0.0130 | 0.0209 |
| ReLU $m =$1,000 | 0.0019 | 0.0042 | 0.0112 | 0.0203 |
| ReLU $m =$10,000 | 0.0018 | 0.0041 | 0.0109 | 0.0203 |
| ReLU $m =$100,000 | 0.0032 | 0.0053 | 0.0111 | 0.0203 |
| Shared Bias | 0.0017 | 0.0040 | 0.0110 | 0.0204 |
| Explain Away | 0.0018 | 0.0041 | 0.0110 | 0.0206 |
| Inhib. Interneuron | 0.0017 | 0.0041 | 0.0109 | 0.0204 |

Table 1: **Reconstruction Validation Accuracy** - The baseline ReLU model varying the number of neurons, compared to the shared bias, explaining away, and inhibitory interneuron networks. We only show noise up to $\sigma = 0.8$ because the higher noise levels present trivial solutions that all networks perform equally well on. Due to very high convergence between random seeds, we just present a single run for each model.

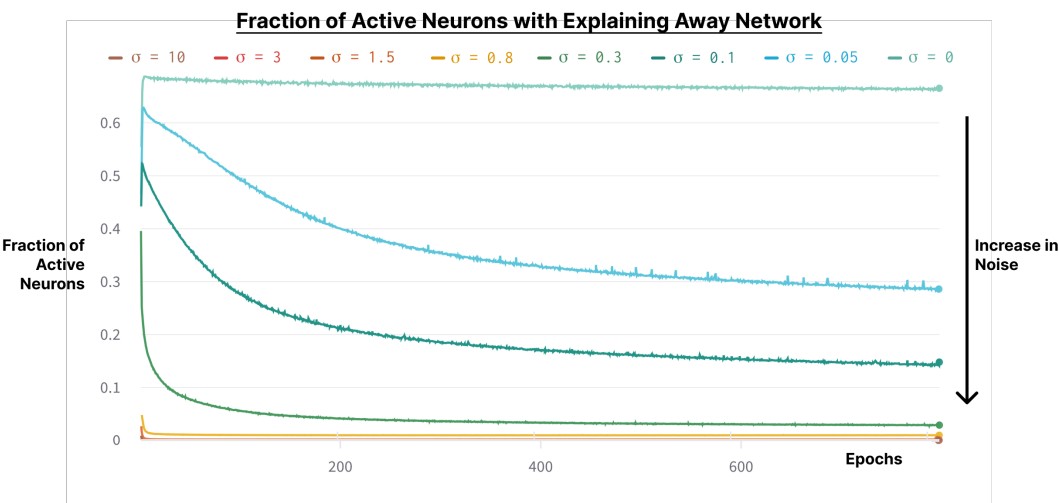

Figure 11: **Explaining Away network is sparser and sooner.** For every level of noise, the explaining away network removes neurons redundantly active and makes them much more sparse.

# D DEAD NEURONS

Here we show the number of dead neurons that are present when we anneal our noise value to 0 for reconstruction (Fig. 12) and classification (Fig. 13).

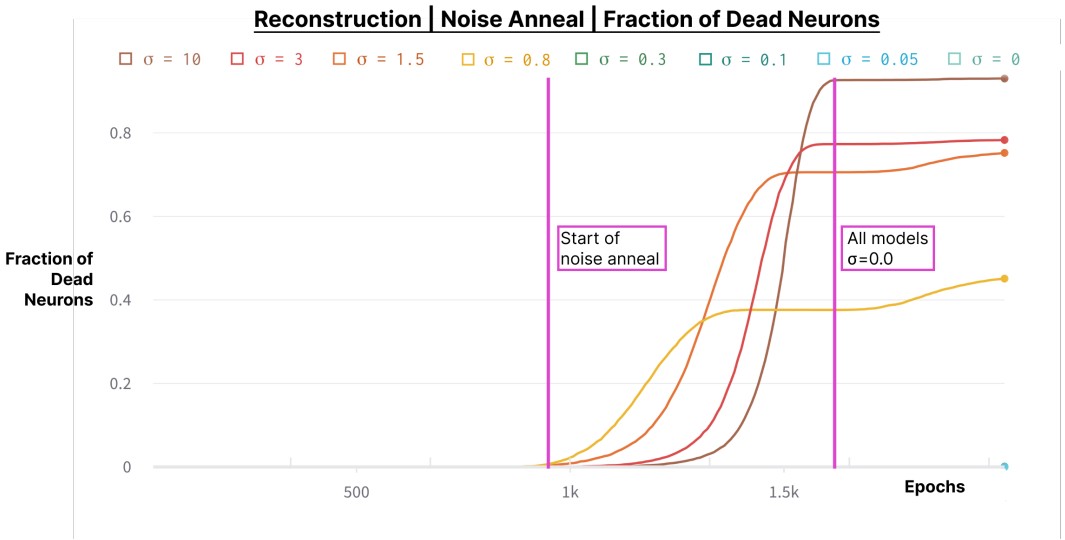

Figure 12: **Dead Neurons during the reconstruction task.** During noise annealing, the higher noise levels $\sigma \geq 0.8$ see a number of dead neurons. However, this does not full explain their sparsity levels and on this trivial task it does not hurt their reconstruction performance.

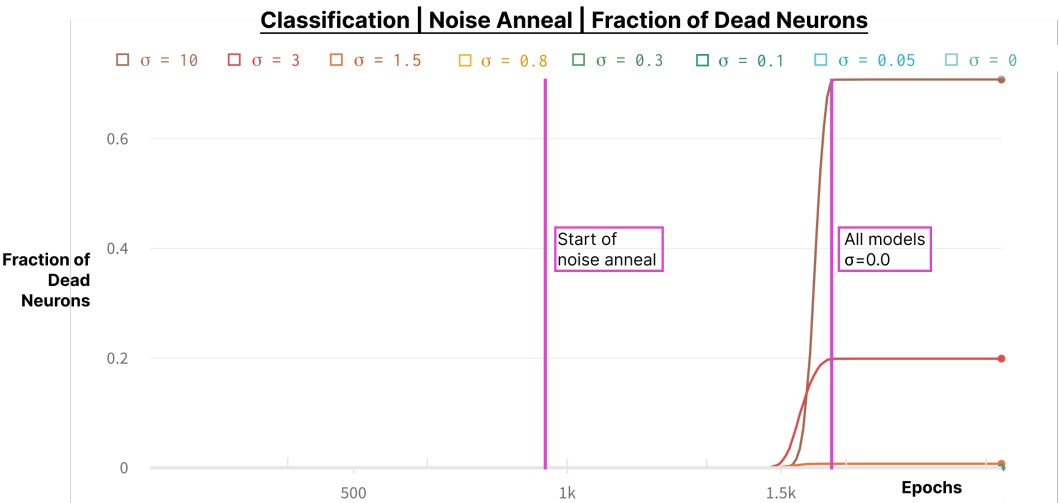

Figure 13: **Dead Neurons during the classification Task.** Here there are fewer dead neurons, they only appear for $\sigma \geq 3.0$ and they only appear close to when the noise is entirely turned off. Again these results do not fully explain the sparsity amounts found.

# E  ACTIVATION FUNCTION ABLATIONS

For CIFAR10, we train each model for 3,000 epochs and present the mean number of active neurons for GELU (Fig. 14) and sigmoid (Fig. 15). We don't show the latent CIFAR10 results as sigmoid and GELU are fully dense with no sparsity when the 0.0001 activation threshold is used. Neither of these networks are as sparse as the ReLU network. Also neither network creates sparsity through dead neurons, but given that all neurons have non-zero absolute activity, having dead neurons is impossible.

While we just show the mean active neurons here, both solutions look like those of Fig. 3 with the neurons for all inputs converging to a Top-K solution where the bias terms become negative and synchronize along with the key vector $L_2$ norms.

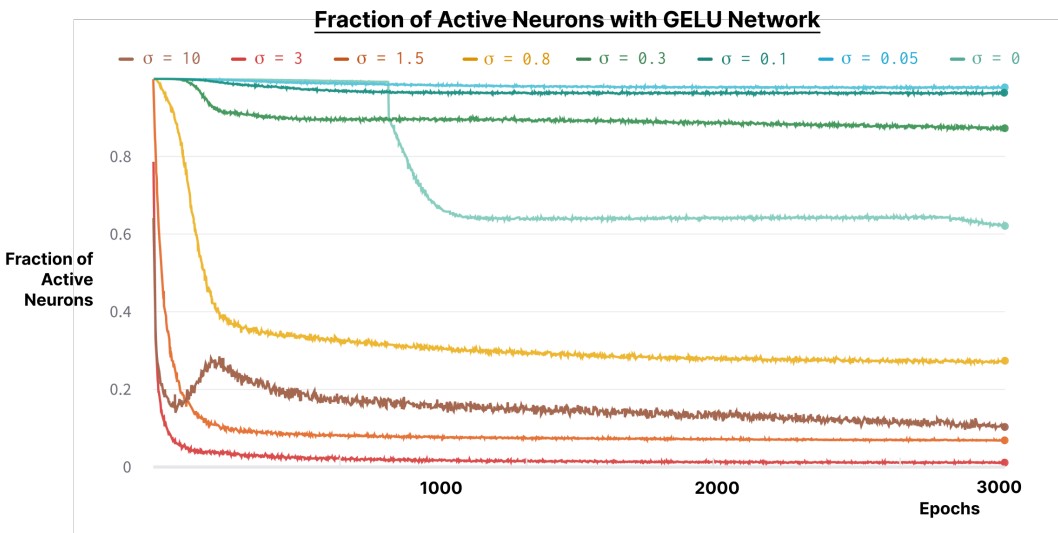

Figure 14: **The GELU network sparsifies with noise when trained on CIFAR10.** The GELU network closely resembles the ReLU network by sparsifying (but not to the same degree). It also forms biological receptive fields. We use the arbitrary 0.0001 absolute activation threshold to label if a neuron is on or off.

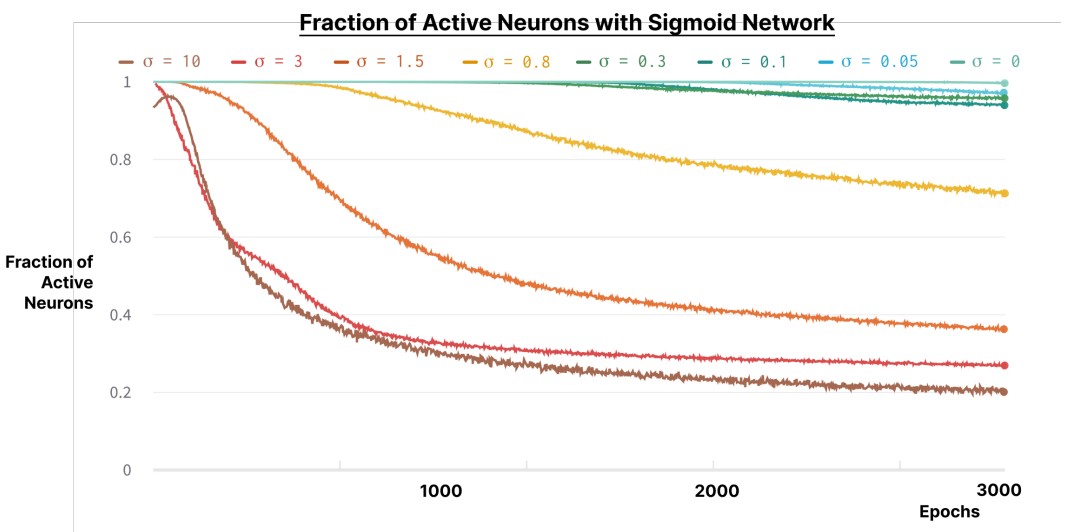

Figure 15: **The sigmoid network also sparsifies with noise for CIFAR10.** While sigmoid sparsifies in proportion to noise, it does not form biological receptive fields and is less sparse than GELU or ReLU.

