# OpenReview forum: "Noise Transforms Feed-Forward Networks into Sparse Coding Networks"
_ICLR.cc/2023/Conference — Submitted to ICLR 2023_

### Official Review · Reviewer_AnUn · 2022-10-25

**Confidence:** 4
**Correctness:** 3
**Technical Novelty And Significance:** 1
**Empirical Novelty And Significance:** 2
**Recommendation:** 3

**Clarity, Quality, Novelty And Reproducibility:**

I am not aware of prior studies showing that noise can induce sparsity.

The text of the paper is clear overall. I was able to follow what they did.

The authors perform many analyses on single-layer networks, which gives one a sense of the robustness of the results in that setting. However, much less is done with multilayer networks, which is a substantial limitation.

The authors provide code implementing their models and analyses.

When the authors train on the output of CIFAR10 embeddings it is unclear whether the noise is applied to the embeddings or the input. It would be useful to do both and report the results.

I assume that when measuring the sparsity of the network, this is done on testing data without noise?

I wondered if the sparsity results were robust to the scale of the weight initializations?


**Strength And Weaknesses:**

Strengths

The paper is simple and easy to follow.

The sparsity effects shown appear relatively large and robust.

Adding noise is a simple manipulation that can have benefits beyond sparsity, and so there is some appeal to using this manipulation as a way to generate sparsity.

Weaknesses

It is odd that the authors focus so much on a one-layer model. For this finding to be broadly relevant, it is important to test whether this effect occurs in large-scale DNNs and whether networks can generate sparsity while maintaining good performance on real-world tasks. The only thing that is done along these lines is with a single, small GPT2 model.

It appears you need fairly high levels of noise to generate substantial sparsity (e.g., sigma=0.3 to get below 20% activation in Figure 2, which based on Figure 4 is quite a lot of noise). It seems this might limit the practical utility of the method, though it is hard to say in part because experiments are mostly limited to the single-layer training regime.

The link with biological receptive fields is tenuous. There are multiple ways of generating center-surround and oriented/bandpass receptive fields (e.g., whitening for center-surround RFs, sparsity for oriented RFs, CNNs often exhibit similar structure at the first layer). It is not clear this approach provides a better fit to biological data than these alternatives and there is no quantification of the match to biology or model comparison. Moreover, there are so many differences between the models trained here and biological networks, that it is hard to know the relevance of this finding, particularly given that the results can vary with the activation function (e.g., the sigmoid activation function does not generate sparsity) and the optimization algorithm (e.g., SGD with a low learning rate apparently does not generate sparsity).


**Summary Of The Paper:**

This paper demonstrates that adding noise to a single network layer causes the activations in that layer to become sparse when the output is given by a ReLU nonlinearity. They first apply this idea to a one-layer autoencoder and show that the degree of sparsity increases as the magnitude of the noise increases (Figure 2). This sparsity is created by a negative bias that is shared across all units coupled with a constrained L2 norm on the encoding weights. The result is a “Top-K” activation profile in which for a given stimulus only the top K units have non-zero values. The authors report that the presence of noise produces center-surround and oriented receptive fields partially reminiscent of the classic paper from Olshausen & Field (1997) (which they report not being able to replicate). They report that this sparsity is maintained when the noise is gradually annealed to zero and is present to a lesser degree for a classification task on the CIFAR dataset.

**Summary Of The Review:**

The paper reports a phenomenon that is novel to knowledge. The relevance to multilayer networks is unclear which limits the impact of the study to the ML community. The relevance to biology is also unclear for the reasons noted above. As a consequence, I am doubtful that the study will have a substantial impact on the ML or neuroscience community.

Response to rebuttal

I apologize for my slow response and thank the authors for responding to each of my points. I still find the paper underwhelming, because of the limitation to shallow models (though I appreciate the authors are working on extending to deeper models) and the lack of any quantitative model comparisons with biological data (which is of course challenging, but needed if one wishes to make claims about relevance to biology).

---

> ### Author Response · Authors · 2022-11-19
> **Reply to Reviewer - First**
>
> We thank the reviewer for their time, comments, and feedback. Replying to each comment in turn.
>
> - **"... partially reminiscent of the classic paper from Olshausen & Field (1997) (which they report not being able to replicate)"**
>
> We do not claim to be unable to replicate this paper and there are a number of differences between them including: not using an MLP, not using ReLU, training on small image patches rather than full images, and implementing both L1 regularization *and* the explaining away circuit (Appendix C). However, we will do a more thorough replication and comparison between our work and traditional sparse coding networks.
>
> - **"It is odd that the authors focus so much on a one-layer model. For this finding to be broadly relevant, it is important to test whether this effect occurs in large-scale DNNs and whether networks can generate sparsity while maintaining good performance on real-world tasks. The only thing that is done along these lines is with a single, small GPT2 model."**
>
> We will test our finding on deeper models and propose the AlexNet and ConvMixer architectures trained on ImageNet. Are there other models that come to mind we should prioritize testing?
>
> - **"It appears you need fairly high levels of noise to generate substantial sparsity (e.g., sigma=0.3 to get below 20% activation in Figure 2, which based on Figure 4 is quite a lot of noise)."**
>
> While the noise levels are large, after annealing the noise we found there was no penalty to model accuracy on the classification task or Transformer next token prediction (in fact slightly higher validation accuracy). We will test this finding on deeper the image classificaiton models proposed.
>
>
> - **"The link with biological receptive fields is tenuous. There are multiple ways of generating center-surround and oriented/bandpass receptive fields (e.g., whitening for center-surround RFs, sparsity for oriented RFs, CNNs often exhibit similar structure at the first layer). It is not clear this approach provides a better fit to biological data than these alternatives and there is no quantification of the match to biology or model comparison. Moreover, there are so many differences between the models trained here and biological networks, that it is hard to know the relevance of this finding, particularly given that the results can vary with the activation function (e.g., the sigmoid activation function does not generate sparsity) and the optimization algorithm (e.g., SGD with a low learning rate apparently does not generate sparsity)."**
>
> What is striking about our receptive fields is not *how* similar they look to biological receptive fields or how much more/less than other models but that they emerge at all just from the introduction of noise.
>
> However, we would be interested in doing a rigorous quantitative way to compare receptive fields across models. Does the reviewer know of any work/techniquest that accomplish this?
>
> With regards to our model being very different from a biological system, this is true and if anything further accentuates that it is striking the network will learn a sparse coding solution from noise alone.
>
> On the sigmoid activation not generating sparsity, our results are more nuanced and we will try to convey them more clearly. Only ReLU produces sparsity across datasets. Both sigmoid and GELU are sparse for the CIFAR10 raw pixel dataset in proportion to noise but dense for the latent CIFAR10 embeddings. While we believe these ablations are useful, they are also not particularly meaningful because ReLU is the only truly sparse activation function that maps activations to 0. Sigmoid and GELU both required arbitrary activation thresholds, for example of 0.001 that we then label as "off".
>
> Our result that SGD with a low enough learning rate did not produce sparse solutions is surprising. We believe it may have to do with this recent work: "SGD with large step sizes learns sparse features" https://arxiv.org/abs/2210.05337. And will run further replications and tests of this finding.

---

> > ### Author Response · Authors · 2022-11-19
> > **Reply to Reviewer - Second**
> >
> > - **"When the authors train on the output of CIFAR10 embeddings it is unclear whether the noise is applied to the embeddings or the input. It would be useful to do both and report the results."**
> >
> > We do both, Figure 7 top right panel shows both the raw CIFAR10 pixels and the CIFAR10 embeddings. We will make this more clear for readers.
> >
> > - **"I assume that when measuring the sparsity of the network, this is done on testing data without noise?"**
> >
> > The results presented are with training data and with the amount of noise given for each model that is either constant throughout training or annealed over time to 0 noise (Fig. 6). Test data gives similar results but we will make this finding more explicit.
> >
> > - **"I wondered if the sparsity results were robust to the scale of the weight initializations?"**
> >
> > Analyzing the L2 norms of our weights in Fig. 3 predicts that the final sparsity amount of each model will be independent of initialization scale. However, the time taken to reach this sparsity level may differ.
> >
> > We cover this analysis in the middle paragraph of pg. 4 starting with *"As noise increases, the..."*. This is where for the high noise setting, the model must wait for its weight norms to become small enough and its bias terms to become negative enough to actually introduce sparsity. However, we will test this theory directly by varying weight init scales across noise settings.

---

### Official Review · Reviewer_wvBd · 2022-10-25

**Confidence:** 4
**Correctness:** 4
**Technical Novelty And Significance:** 2
**Empirical Novelty And Significance:** 2
**Recommendation:** 3

**Clarity, Quality, Novelty And Reproducibility:**

The present work adds to the literature on the effects of adding noise in neural models. However, the advance is minor, and easily realized, as the authors observe, by imposing a negative bias to the neurons instead.  The communication of the methods and results are clear, and the code to reproduce the results has been shared as well (note: this code should be listed in a separate "Reproducibility" section before the references if possible).

**Strength And Weaknesses:**

Strengths

1. The authors find a simple, implicit manner to create top-k networks that compares favorably to explicit methods

2. The method is tested in several different circumstances, with different tasks, numbers of neurons, and noise levels. Simple characteristics of the network, such as the fraction of active neurons and their biases, are well described.

Weaknesses

Major

1. As the authors observe, the results are recapitulated using a global negative bias term, acting as common inhibition that requires greater input values to activate neurons and resulting in top-k networks.  Thus, while such an observation may indeed be useful for those interested in such networks, the advance it comprises is a minor one.

Minor

1. While the presented results may bear on biological considerations, as suggested therein, caution is advised, as noise in biological networks is dynamic and thus not perfectly analogous to simply adding noise to inputs of a feedforward network.

**Summary Of The Paper:**

The present work explores the relationship between input noise and network sparseness.  Such sparseness is known to exist in biological networks (brains), and has been hypothesized to arise as a solution to increase the signal-to-noise ratio of the network and to keep energy consumption low. Here, the authors find that adding noise to ANNs effectively decreases the biases of the constituent neurons, resulting in only the top k neurons being active. The authors further find that such noise results in receptive fields that are similar to those found in V1.  The results suggest that sparse neuronal activity in brains may arise to better deal with noisy inputs rather than (or in addition to) minimizing energy consumption.  Such sparse, top-k networks may also be useful in future, low-energy ANN applications.

**Summary Of The Review:**

While the authors demonstrate that injecting noise into feedforward networks sparsifies their responses, such results can easily be explained and recapitulated--and as noted by the authors--by simply globally reducing the neurons' biases.  Thus the presented results comprise a minor advance in the area

---

> ### Author Response · Authors · 2022-11-19
> **Reply to Reviewer**
>
> We thank the reviewer for their time, comments, and feedback. Replying to each comment in turn.
>
> - **"As the authors observe, the results are recapitulated using a global negative bias term, acting as common inhibition that requires greater input values to activate neurons and resulting in top-k networks. Thus, while such an observation may indeed be useful for those interested in such networks, the advance it comprises is a minor one."**
>
> What is striking about this finding is not the complexity of it, which we agree is straightforward, but that the network *wants* to be sparse.
>
> While it depends on the task and architecture, as shown by our zero noise setting neural networks empirically don't become highly sparse on their own. The other common routes to induce sparsity are L1 regularization of activations and enforcing a Top-K activation function. We find that neither of these approaches is as stable and successful as noise in encouraging the network to find a sparse solution.
>
> We will re-write parts of our introduction and results to better contextualize this result.
>
> - **"While the presented results may bear on biological considerations, as suggested therein, caution is advised, as noise in biological networks is dynamic and thus not perfectly analogous to simply adding noise to inputs of a feedforward network."**
>
> We agree that *some* noise in biological systems is dynamic and correlated with neural activity. However, there is also uncorrelated random noise in any biological process. For example, Brownian noise that influences the diffusion neurotransmitters. Principles of Neural Design (Peter Sterling and Simon Laughlin) gives many such examples across brain regions.
>
> Moreover, for the inputs trained on directly on image pixels, Gaussian and Poisson noise is realistic for modeling photon noise on photoreceptors.
>
> Nevertheless, we will add this as a limitation of our work for readers to keep in mind.

---

### Official Review · Reviewer_39gB · 2022-10-26

**Confidence:** 2
**Correctness:** 3
**Technical Novelty And Significance:** 2
**Empirical Novelty And Significance:** 2
**Recommendation:** 3

**Clarity, Quality, Novelty And Reproducibility:**

Clarity:
* It was difficult to follow what  precisely is meant by "Top-K" networks. Adding a more rigorous definition to the introduction will benefit the paper.

Novelty:
* The fact that a 1-hidden layer denoising auto-encoder learns "sparse-coding"-like features is well-known.  See e.g., Figure 2 in the paper "Marginalized Denoising Auto-encoders for Nonlinear Representations" .

Reproducibility:
* The results in the paper seem reproducible thanks to the provided code.

**Strength And Weaknesses:**

Strengths:
* The paper tackles the important problem of bridging the gap between biological plausible learning and current artificial neural networks.

Weaknesses:
* The paper mostly focuses on 1-hidden layer networks. It is not too clear why and how these results are relevant for practice, where typically much deeper networks are considered. For classification tasks, the observed sparsity seems to be much less pronounced.
* Unlike claimed in the paper, using the L1-norm can lead to biologically plausible and sparse features even for classification tasks, see e.g. Figure 4 in "Towards Learning Convolutions from Scratch" (https://papers.nips.cc/paper/2020/file/5c528e25e1fdeaf9d8160dc24dbf4d60-Paper.pdf)
* There are some issues regarding novelty/clarity (see next section)


**Summary Of The Paper:**

The paper shows that when adding noise to the input, sparse and "biologically plausible" features emerge in a 1-hidden layer autoencoder. Moreover, using the ReLU-activation function seems to important for this pattern to emerge.

**Summary Of The Review:**

Overall, the paper tackles an important problem but there are some issues regarding novelity, clarity and some errornous claims. Therefore, I cannot recommend acceptance of the paper at this stage.

---

> ### Author Response · Authors · 2022-11-19
> **Reply to Reviewer**
>
> We thank the reviewer for their time, comments, and feedback. Replying to each comment in turn.
>
> - **"The paper mostly focuses on 1-hidden layer networks. It is not too clear why and how these results are relevant for practice, where typically much deeper networks are considered. For classification tasks, the observed sparsity seems to be much less pronounced."**
>
> We propose testing our model with AlexNet and ConvMixer architectures to analyze sparsity and validation accuracy effects. Are there additional models the reviewer suggests we test?
>
> - **"Unlike claimed in the paper, using the L1-norm can lead to biologically plausible and sparse features even for classification tasks, see e.g. Figure 4 in "Towards Learning Convolutions from Scratch" (https://papers.nips.cc/paper/2020/file/5c528e25e1fdeaf9d8160dc24dbf4d60-Paper.pdf)"**
>
> The referenced paper uses L1 regularization on the model *weights* not *activations*. We make no claims in this work about weight sparsity and view the two as related (weight sparsity can increase activation sparsity) but fundamentally different and out of the domain of sparse coding.
>
> We will make this discrepancy clearer in our paper.
>
> - **"It was difficult to follow what precisely is meant by "Top-K" networks. Adding a more rigorous definition to the introduction will benefit the paper."**
>
> Thanks, we agree and will add a clear definition of Top-K.
>
> - **"The fact that a 1-hidden layer denoising auto-encoder learns "sparse-coding"-like features is well-known. See e.g., Figure 2 in the paper "Marginalized Denoising Auto-encoders for Nonlinear Representations" ."**
>
> We will be sure to cite this paper. However, the result presented is from the highly artificial MNIST dataset rather than naturalistic images. With the MNIST digits being composed of strokes, the Gabor like filters that are learnt can be interpreted as being stroke detectors and a consequence of this particular dataset. This is not the case for the natural CIFAR10 images making our finding more general and biologically relevant. We will be sure to make this difference more explicit.

---

### Official Review · Reviewer_6F1u · 2022-11-03

**Confidence:** 4
**Correctness:** 3
**Technical Novelty And Significance:** 1
**Empirical Novelty And Significance:** 3
**Recommendation:** 3

**Clarity, Quality, Novelty And Reproducibility:**

The paper is clearly written.
About novelty, the fact that Denoising Auto-Encoders produces Gabor-like filter has already been observed in previous work (e.g. in [1])
Reproducibility is allowed thanks to a github repository.

[1] : Extracting and composing robust features with denoising autoencoders, P. Vincent et. al. ICML 2008

**Strength And Weaknesses:**

The principal strength of this paper is that the set of experiments presented in this paper is interesting and is worth sharing with the community.

The main weakness of the paper is the fact that the focus is on observing the obtained sparsity. Reconstruction accuracies are only compared using a top-K network. I think that comparing with an actual sparse coding model is important to control for the quality of the solution learned.

Similarly, about the classification accuracy comparison, it would be important to report the accuracies for all noise levels as well as all number of hidden units and compare with an equivalent sparse coding representation.





**Summary Of The Paper:**

In this empirical paper, the authors observe the correlation between the sparsity of the learned representations and the level of noise injected into the input of Neural Networks with the ReLU activation function. Experiments have been carried out on CIFAR10 both on pixel data and on a neural representation, MNIST, and on the WikiText-103 dataset. They report a positive correlation between the magnitude of the injected noise and the sparsity of the representation. They observe Gabor-like filters when reconstructing with noise on pixel data. For the classification task, the performance of the model did not decrease with sparsity. Also, compared with using only the top-K activations of a model learned without noise, the models learned with noise gave better validation accuracies.

**Summary Of The Review:**

While I think that the experiments presented are interesting, I think that adding baselines to control for the quality of the sparse representation obtained is necessary to allow for acceptance of the work.

---

> ### Author Response · Authors · 2022-11-19
> **Reply to Reviewer**
>
> We thank the reviewer for their time, comments, and feedback. Replying to each comment in turn.
>
> - **"The main weakness of the paper is the fact that the focus is on observing the obtained sparsity. Reconstruction accuracies are only compared using a top-K network. I think that comparing with an actual sparse coding model is important to control for the quality of the solution learned. Similarly, about the classification accuracy comparison, it would be important to report the accuracies for all noise levels as well as all number of hidden units and compare with an equivalent sparse coding representation."**
>
> This is a good point, we will run comparisons to a full sparse coding model and compare them.
>
> - **"The paper is clearly written. About novelty, the fact that Denoising Auto-Encoders produces Gabor-like filter has already been observed in previous work (e.g. in [1])"**
>
> Thank you for this reference that we will cite. While you are correct that Figure 3 in the reference shows what appear to be Gabor like filters, these are on the MNIST dataset rather than naturalistic images. The fact the digits consist of strokes makes it easy to interpret the receptive fields as being stroke detectors for this specific dataset rather than generalized Gabor filters that appear in human V1 and our model with CIFAR10. We will be sure to emphasize this difference.

---

### Decision · Program_Chairs · 2023-01-20

**Decision:**

Reject

**Justification For Why Not Higher Score:**

Reviewers were unanimous in their decision, with which I agree 100% after reading the paper and reviews. Furthermore, the rebuttal did not significantly change the main points of criticism.

**Justification For Why Not Lower Score:**

N/A

**Metareview: Summary, Strengths And Weaknesses:**

Reviewers agree that while interesting, the paper is not ready for publication. Several major issues were raised with respect to missing baselines (sparse coding), limited novelty (Denoising Auto-Encoders and related RBMs are well known to yield Gabor-like filters [1,2]) and weak experimental results (small models, datasets, and no clear sign that the proposed method can yield sparsity while retaining performance on tasks and models of interest).

Perhaps even more important are the points raised by [AnUn]. There are many ways to induce sparse activations and Gabor-like filters. Unfortunately for ML practitioners, the authors do not convincingly establish why their method should be preferred or why it should be integrated into ML pipelines. On the other hand, the paper will also be of limited interest to the neuroscience community as no effort is made to establish fit to biological data. I would encourage the authors to pursue one of these directions before resubmitting.

[1] https://www.jmlr.org/papers/volume11/vincent10a/vincent10a.pdf, see Fig 6 for DAEs applied to natural image patches.
[2] https://www.cs.toronto.edu/~fritz/absps/reluICML.pdf (RBMs applied to natural images of faces, Fig 8)